# A Novel Pathogenic Variant in *MT-CO2* Causes an Isolated Mitochondrial Complex IV Deficiency and Late-Onset Cerebellar Ataxia

**DOI:** 10.3390/jcm8060789

**Published:** 2019-06-04

**Authors:** Charlotte M. Zierz, Karen Baty, Emma L. Blakely, Sila Hopton, Gavin Falkous, Andrew M. Schaefer, Marios Hadjivassiliou, Ptolemaios G. Sarrigiannis, Yi Shiau Ng, Robert W. Taylor

**Affiliations:** 1Wellcome Centre for Mitochondrial Research, Institute of Neuroscience, Newcastle University, Newcastle upon Tyne NE2 4HH, UK; CharlotteZierz@web.de (C.M.Z.); karen.baty@nuth.nhs.uk (K.B.); emma.watson@nuth.nhs.uk (E.L.B.); sila.hopton@ncl.ac.uk (S.H.); gavin.falkous@nuth.nhs.uk (G.F.); andrew.schaefer@nuth.nhs.uk (A.M.S.); robert.taylor@ncl.ac.uk (R.W.T.); 2NHS Highly Specialised Mitochondrial Diagnostic Laboratory, Newcastle upon Tyne Hospitals NHS Foundation Trust, Newcastle upon Tyne NE2 4HH, UK; 3Department of Neurology, Royal Victoria Infirmary, Newcastle upon Tyne Hospitals NHS Foundation Trust, Newcastle upon Tyne NE1 4LP, UK; 4Academic Directorate of Neurosciences, Sheffield Teaching Hospitals NHS Foundation Trust, Royal Hallamshire Hospital, Sheffield S10 2JF, UK; m.hadjivassiliou@sheffield.ac.uk; 5Department of Clinical Neurophysiology, Sheffield Teaching Hospitals NHS Foundation Trust, Royal Hallamshire Hospital, Sheffield S10 2JF, UK; p.sarrigiannis@sheffield.ac.uk

**Keywords:** isolated COX deficiency, cerebellar ataxia, mitochondrial DNA, single fibre segregation studies, heteroplasmy

## Abstract

Both nuclear and mitochondrial DNA defects can cause isolated cytochrome *c* oxidase (COX; complex IV) deficiency, leading to the development of the mitochondrial disease. We report a 52-year-old female patient who presented with a late-onset, progressive cerebellar ataxia, tremor and axonal neuropathy. No family history of neurological disorder was reported. Although her muscle biopsy demonstrated a significant COX deficiency, there was no clinical and electromyographical evidence of myopathy. Electrophysiological studies identified low frequency sinusoidal postural tremor at 3 Hz, corroborating the clinical finding of cerebellar dysfunction. Complete sequencing of the mitochondrial DNA genome in muscle identified a novel *MT-CO2* variant, m.8163A>G predicting p.(Tyr193Cys). We present several lines of evidence, in proving the pathogenicity of this heteroplasmic mitochondrial DNA variant, as the cause of her clinical presentation. Our findings serve as an important reminder that full mitochondrial DNA analysis should be included in the diagnostic pipeline for investigating individuals with spinocerebellar ataxia.

## 1. Introduction

Cytochrome *c* oxidase (COX; also known as complex IV) is the terminal component of the mitochondrial respiratory chain, receiving electrons from cytochrome *c* to reduce molecular oxygen, forming water, which is coupled to the pumping of protons across the mitochondrial inner membrane. Complex IV consists 14 subunits, of which three essential subunits (MT-CO1, MT-CO2 and MT-CO3) form the catalytic core of the enzyme and are encoded by mitochondrial DNA (mtDNA); the remaining eleven subunits are all encoded by nuclear genes [1]. Intriguingly, mitochondrial disorders related to isolated COX deficiency are most commonly associated with nuclear defects that affect the assembly and biogenesis of complex IV, including recessively-inherited pathogenic variants in *SURF1* [2], *SCO1* [3] and *SCO2* [4].

Primary pathogenic variants in the mtDNA-encoded COX subunits are, by comparison, relatively rare [5,6], and are associated with variable clinical manifestations including recurrent rhabdomyolysis [7,8], myopathy and lactic acidosis [9,10], encephalomyopathy [11], adult-onset Leigh syndrome [12], and mitochondrial encephalomyopathy, stroke-like episode and lactic acidosis (MELAS) [13].

We report an adult patient who presented with a late-onset progressive cerebellar ataxia and mild hearing loss at the age of 52 years. A diagnostic skeletal muscle biopsy revealed marked COX deficiency prompting full mtDNA genome sequencing which identified a novel, heteroplasmic variant (m.8163A>G predicting p.(Tyr193Cys); GenBank accession number NC_01920.1) in the *MT-CO2* gene.

## 2. Methods

### 2.1. Case Report

Our patient presented to the Sheffield Ataxia Centre at the age of 52. She noticed dysarthria in her mid-40s and became unsteady on her feet around six-months later. The patient attributed the onset to psychological trauma experienced at the time. Her balance impairment and gait difficulty progressed gradually leading to frequent falls, and she sustained bony injuries including wrist and ankle fractures on a few occasions. She also complained of incoordination in the upper limbs. She had both head and upper limb tremor, which could be exacerbated by emotion. She experienced mild swallowing difficulties, but there was no report of nasal regurgitation, choking episode or aspiration pneumonia. She complained of paraesthesia in her hands and feet. There was an asymmetrical hearing deficit that occurred in her early 50s. She lost a significant amount of weight (dropping from size 12 to size 8) over several months. There was no history of seizure, ptosis, muscle weakness, cardiac problem or diabetes mellitus. She was a non-smoker and rarely consumed alcohol. Her mother died from a stroke in her 60s and her father died from myocardial infarction at the age of 61 years. She had a brother who had ischaemic heart disease. Her son was fit and well.

Clinical examination at presentation showed head titubation, truncal tremor, dysarthria, bilateral intention tremor and pass-pointing, and dystonic posturing of fingers. Her gait was unsteady but not broad-based. She had high arch feet that were suggestive of pes cavus, and hammer toes. There was no ptosis, ophthalmoplegia or reduced muscle power. She had brisk reflexes throughout, except both the ankle jerks were absent. She had bilateral flexor plantar response. Sensory testing identified a loss of vibration sense up to the knee level in the lower limbs.

Routine laboratory investigations including blood count, kidney and liver functions, HbA1c, creatinine kinase level and resting serum lactate were normal except borderline primary hyperthyroidism was identified (TSH 0.26, range 0.3–4.7 mU/L; free thyroxine 23.4, range 9.5–21.5 pmol/L; free T3 5.7, range 3.5–6.5 pmol/L). Her resting 12-lead electrocardiogram and echocardiography were normal. Cranial magnetic resonance imaging (MRI)showed severe cerebral (particularly affecting occipital, parietal and frontal lobes) and cerebellar atrophy (Figure 1); there was no signal abnormality in the basal ganglia. There was some confluent periventricular T2 signal symmetrical change. Nerve conduction studies and electromyography (EMG) identified changes that were consistent with a moderate large fiber axonal sensorimotor neuropathy with features suggesting a length-dependent distribution. However, there was no evidence of myopathy.

Our patient was tested negative for common spinocerebellar ataxia (SCA) 1, 2, 3, 6, 7 and Friedreich’s ataxia. A diagnostic skeletal muscle biopsy (right deltoid) was performed.

### 2.2. Electrophysiological Studies

The patient was referred for electrophysiological assessment to assess the possibility of cortical myoclonus. The Natus Quantum amplifier (Optima Medical Ltd, Guildford, Surrey, UK) at a sampling rate of 2048 Hz was used for the electroecenphalography (EEG)/EMG polygraphy examination (analogue bandwidth 0.01–680 Hz). Spike 2 (version 8.12) software (CED Ltd, Cambridge, UK) was used for further quantitative EEG/EMG data analysis. The in-built Spike 2 software functions for cross-correlation and Fast Fourier transform analysis were used. Somatosensory evoked potential (SEP) recordings were also performed based on published recommendations [14] and the possibility of C-reflexes was assessed with surface EMG electrodes on upper and lower limbs according to the methodology and normal values described in previous work [15].

### 2.3. Histopathological Analysis of the Muscle Biopsy

Standard histology (hematoxylin and eosin (H&E)) and oxidative enzyme histochemistry (cytochrome c oxidase (COX), succinate dehydrogenase (SDH) and sequential COX/SDH activities) were performed on 10 µm transversely-oriented frozen skeletal muscle sections as described previously [16]. Quantitative, quadruple immunofluorescence was used to interrogate NDUFB8 (complex I) and COXI (complex IV) immunoreactivities and was performed exactly as reported [17].

### 2.4. Mitochondrial Genetic Studies

The entire mitochondrial genome was amplified using two overlapping long polymerase chain reaction (PCR) amplicons, NC_012920.1:m.550-9839 (amplified using oligonucleotide primers m.550-569 and m.9839-9819) and NC_012920.1:m.9592-645 (amplified using oligonucleotide primers m.9592-9611 and m.645-626), pooled in equimolar amounts after quantification using Ion Library Taqman™ Quantitation kit (Thermo Fisher Scientific, Paisley, UK) and fragmented using Ion Shear™ Plus (Thermo Fisher Scientific, UK), with barcodes and adaptors ligated using the Ion Xpress™ Plus Fragment Library Kit (Thermo Fisher Scientific, Paisley, UK). Automated template preparation (200 bp library), chip loading and sequencing were performed using the Ion Torrent PGM (Thermo Fisher Scientific, UK) in conjunction with the Ion Torrent Ion Chef system (v5.8.0) (Thermo Fisher Scientific, UK). Data were aligned to the revised Cambridge reference sequence for human mtDNA (GenBank Accession number: NC_012920.1) and analysis performed using Torrent Suite v5.8.0 (Thermo Fisher Scientific, UK) using Variant Caller v5.8.0.19 (custom settings) and Coverage Analysis v5.8.0.8 plugins. The mitochondrial genome was covered at a minimum read depth of 200x. Analytical sensitivity for single nucleotide variants present at ≥5% heteroplasmy, was ≥95% (95% confidence interval).

### 2.5. Single Fibre Segregation Analysis and mtDNA Heteroplasmy Assessment

A novel m.8163A>G variant identified following mtDNA sequencing was further assessed in patient tissues and individual (COX-deficient and COX-positive) skeletal muscle fibres isolated by laser-capture microdissection by quantitative pyrosequencing (Pyromark Q24 platform, Qiagen, Manchester, UK) using mutation-specific primers (details available on request). The allele quantification application of Pyromark’s proprietary Q24 software was used to calculate heteroplasmy levels (level of test sensitivity >3% mutant mtDNA) [18].

### 2.6. Western Blotting and Blue Native PAGE in Patient Muscle

Protein extraction from patient skeletal muscle for SDS-PAGE (polyacrylamide gel electrophoresis) and subsequent western blotting was carried out as described previously [19]. Blue Native PAGE was conducted on mitochondria isolated from skeletal muscle samples as described previously [20] using antibodies against COXI (abcam ab14705), COXII (abcam ab110258), SDHA (abcam ab14715), Porin/VDAC1 (abcam ab14734), UQCRC2 (abcam ab14745), NDUFB8 (abcam ab110242), ATP5A (abcam ab14748) and ATP5B (abcam ab14730). All primary antibodies were used at a dilution of 1 in 1000, except for SDHA, which was diluted at 1 in 2000.

## 3. Results

### 3.1. Electrophysiological Findings

Polygraphy recordings captured on surface EMG channels from the left arm indicated a low frequency sinusoidal postural tremor at 3 Hz (Figure 2), which is below the frequency of most pathologic tremors that lies between 4 to 12 Hz [21]. Contrary to what can be expected in myoclonus, where typically co-activation of agonist/antagonist with small duration EMG discharges, typically below 75 ms in cortical myoclonus [22,23] occurs, we detected an out-of-phase relationship between pairs of antagonists and between proximal and distal muscles. In addition, SEPs showed no evidence of cortical hyperexcitability or any evidence of reflex myoclonus. The electroclinical findings, particularly the low frequency of the tremor, [24] were pointing in the direction of a cerebellar tremor syndrome.

### 3.2. Histochemistry and Immunohistochemical Analyses of Muscle Biopsy

The histochemical activity of succinate dehydrogenase (SDH) was well-preserved throughout the biopsy, whilst both the individual COX and sequential COX/SDH histochemical reactions revealed extensive COX-deficiency, affecting in excess of 90% of all fibres; some clearly COX-positive fibres were apparent (Figure 3A). In support of this, quadruple immunofluorescence analysis confirmed a mitochondrial defect involving complex IV in isolation, a significant loss of COXI protein expression but normal levels of NDUFB8 expression (Figure 3B).

### 3.3. Mitochondrial Genetic Studies

Following a screen of common mtDNA variants and mtDNA rearrangements, sequencing of the complete mitochondrial genome was undertaken. This revealed a novel m.8163A>G (p.(Tyr193Cys)) *MT-CO2* variant at high levels of mtDNA heteroplasmy in the muscle (89% mutant load), affecting a highly-conserved amino acid (Figure 3C). Quantitative pyrosequencing of DNA samples obtained from other tissues (buccal epithelia, urinary sediment and blood sample) revealed lower levels of mtDNA heteroplasmy (49%, 49% and 5% mutant loads, respectively), whilst we were unable to detect the m.8163A>G variant in blood and urine DNA samples from the patient’s clinically-unaffected brother.

Single muscle fibre analysis of individual COX-positive and COX-deficient fibres revealed a statistically significant higher mutation load in COX-deficient fibres (97.1 ± 0.7% (*n* = 7 fibres)) than in COX-positive fibres (30.2 ± 6.2% (*n* = 12 fibres); unpaired *t*-test *p* ≤ 0.0001) confirming pathogenicity of the m.8163A>G variant (Figure 3D).

### 3.4. Steady-State Levels of Respiratory Chain Components and Complexes

The steady-state protein levels of subunits of the mitochondrial oxidative phosphorylation (OXPHOS) complexes in patient muscle and aged-matched control samples were analysed by SDS-PAGE and immunoblotting. We observed the quantitative loss of mitochondrial (COX I and COX II) subunits of Complex IV, whilst the levels of other OXPHOS components were normal in the patient (Figure 3E). This was associated with a failure of the assembly of complex IV holoenzyme as determined by one-dimensional BN-PAGE (Figure 3F).

## 4. Discussion

Our patient has primary mtDNA disease characterised by late-onset, progressive cerebellar syndrome, tremor and mild hearing deficit as well as having brisk reflexes and axonal neuropathy. There was no family history of a progressive neurological disorder although her mother had passed away from a stroke. We believe that the m.8163A>G variant accounts for her clinical phenotype based on several lines of evidence. First, the muscle biopsy showed a marked abnormality in histochemical COX activity, which was mirrored by the isolated loss of COXI protein expression following quadruple OXPHOS immunohistochemistry. Second, full mtDNA genome sequencing excluded all previously reported pathogenic mtDNA variants, identifying the novel m.8163A>G variant as an excellent pathogenic candidate; this was not seen in >1900 mtDNA sequences in our own in-house database nor on publicly-available databases. Third, the m.8163A>G variant was heteroplasmic and present in a post-mitotic muscle at higher levels than replicative cells. It also segregated with clinical disease in the family. Fourth, p.Tyr193 in COXII represents an invariant amino acid in highly conserved domain of the protein, and mutation of this protein would corroborate the observed biochemical defects seen histochemically and immunohistochemically; western blotting and BN-PAGE showed a complete loss of COXII protein and assembly of complex IV holoenzyme, respectively. Finally, single muscle fibre segregation studies confirmed higher levels of m.8163A>G in COX-deficient fibres compared to COX-positive fibres, demonstrating segregation of mutant load with a respiratory chain defect.

Since the first pathogenic *MT-CO2* variant was reported in 1999 [9], thirteen other pathogenic *MT-CO2* variants, including our case, have been identified (Appendix A). Exercise intolerance and lactic acidosis, with [7] or without [9] rhabdomyolysis, were relatively common and described associated with six *MT-CO2* variants. The most devastating presentation was reported in a prematurely born infant who died from apnoea and severe metabolic acidosis due to a de novo, frame-shift mutation (m. 8042delAT) [25]. It is intriguing that the phenotype-expressing threshold was very low associated with this particular *MT-CO2* variant, with mutant heteroplasmy levels identified at <20% in five different tissues including skeletal and cardiac muscles. Classical syndromes associated with mitochondrial diseases, such as MELAS [26], LHON [27] and Alper-like syndrome [28], were rare. There are some similarities in terms of the clinical picture between the current case and another pathogenic variant (m.7587T>C) that was previously reported by our laboratory [11]. The proband presented with a childhood-onset, progressive cerebellar ataxia, cognitive impairment and retinitis pigmentosa, whilst his mother only exhibited mild cerebellar ataxia at the age of 47 years. However, pyramidal features were absent, in contrast to the present case with the m.8163A>G, p.(Tyr193Cys) variant.

Our case also serves to illustrate the marked phenotypic heterogeneity associated with primary mtDNA disease. She had a progressive cerebellar syndrome as the predominant clinical finding whilst typical features of adult mitochondrial disease, such as maternal inheritance, myopathic symptoms and multi-system abnormalities, are absent. The apparent negative family history and indolent cerebellar ataxia in our patient could have been mistaken as an autosomal recessive disorder and subjected to large panel gene screening as recommended in current clinical practice [29,30,31], without identifying the causal mtDNA variant. The role of a diagnostic muscle biopsy in reaching securing a definitive diagnosis in our case is conspicuous. Indeed, the mitochondrial disease has recently been identified as the fourth most common cause of genetic-determined ataxia [32], and muscle biopsy continues to play a pivotal role in diagnostic algorithms [33].

In addition, despite cortical or subcortical myoclonus being a classical feature of the mitochondrial disease [34], a low frequency, mainly postural 3–4 Hz tremor was documented in this case.

In summary, we recommend that muscle biopsy and full mitochondrial genome sequencing should be considered as part of the investigation for the adult-onset progressive cerebellar syndromes after excluding common acquired and genetic aetiologies, given de novo, rare pathogenic mtDNA pathogenic variants may present with atypical mitochondrial syndromes, as nicely illustrated by this case.

## Figures and Tables

**Figure 1 jcm-08-00789-f001:**
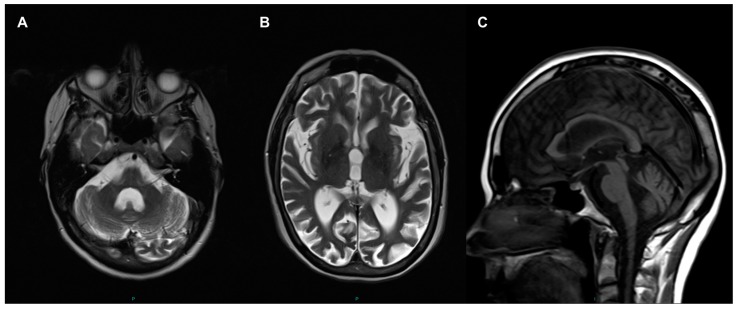
Cranial magnetic resonance imaging (MRI) performed at age 46 years. Enlarged 4th ventricle and hemispheric atrophy of cerebellum (**A**); and dilated ventricles and generalised cerebral atrophy shown in axial T2 (**B**); Sagittal T1 view shows cerebellar atrophy and normal brainstem (**C**).

**Figure 2 jcm-08-00789-f002:**
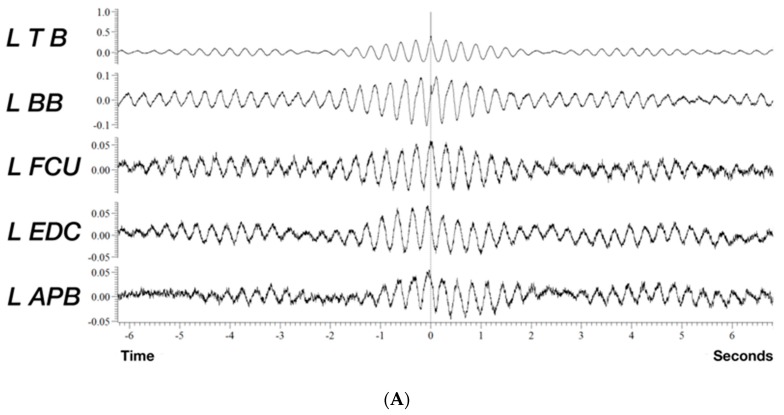
Quantitative analysis of a 90 s epoch of tremorogenic surface EMG recordings from the left arm whilst the patient maintains anti-gravity posture. (**A**) Cross-correlation analysis using the triceps brachii EMG channel as reference reveals a mainly out of phase relationship between the biceps and triceps brachii and a similar out of phase relationship between agonist/antagonist in the forearm. (**B**) Fast Fourier transform (FFT) shows peak frequency at 3.3 Hz and the first harmonic of the tremor at 6.6 Hz (FFT resolution 0.1 Hz). APB: abductor pollicis brevis, EDC: extensor digitorum communis, FCU: flexor carpi ulnaris, TB: triceps brachii, BB: biceps brachii.

**Figure 3 jcm-08-00789-f003:**
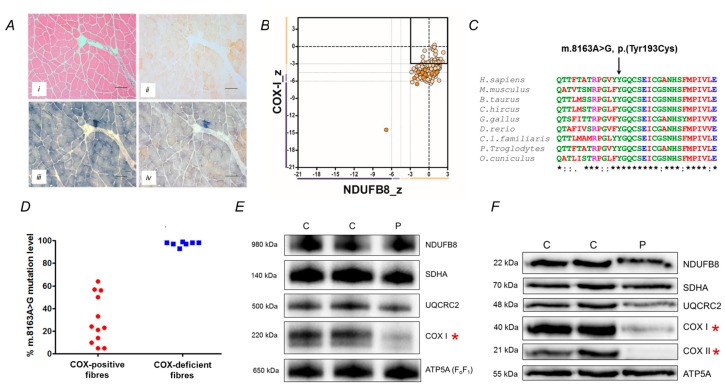
Histopathological, biochemical and molecular characterisation of a novel. *MT-CO2* variant. (**A**) Histological and histochemical analyses of the patient’s skeletal muscle biopsy showing (i) hematoxylin and eosin (H&E) staining, (ii) COX, (iii) SDH and (iv) sequential COX-SDH histochemistry; scale bar, 100 µm. (**B**) Quadruple immunofluorescence analysis of NDUFB8 (complex I) and COXI (complex IV) showing a marked loss of COXI expression. Each dot represents the measurement from an individual muscle fibre, color co-ordinated according to its mitochondrial mass (low = blue, normal = beige, high = orange, very high = red). Gray dashed lines represent standard deviation (SD) limits for a classification of the fibres. Lines next to x- and y-axis represent the levels of NDUFB8 and COXI: beige = normal (>−3), light beige = intermediate positive (−3 to −4.5), light purple = intermediate negative (−4.5 to −6), purple = deficient (<−6). Bold dashed lines represent the mean expression level of normal fibres. (**C**) Multiple sequence alignment highlighting the evolutionary conservation of p.(Tyr193) which is mutated in the m.8163A>G patient. (**D**) Single fibre PCR analysis clearly shows marked segregation of the m.8163A>G variant with a biochemical defect in individual COX-deficient muscle fibres. (**E**) Muscle protein from age-matched controls (C) and patient (P) muscle (15 µg) were analysed by SDS PAGE (12%) and immunoblotting (see Methods), revealing a decrease in steady-state levels of both COX1 and COX2 (highlighted by asterisk) (**F**) Mitochondrial proteins (4.5 µg) isolated from patient muscle (P) and controls (C) and analysed by one dimensional BN-PAGE (4 to 16% gradient) using subunit-specific oxidative phosphorylation system (OXPHOS) antibodies show decreased assembly of complex IV (asterisked). Complex II (SDHA) was used as a loading control in both (panels **E** and **F**).

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
