# Peer review of "A Novel Pathogenic Variant in MT-CO2 Causes an Isolated Mitochondrial Complex IV Deficiency and Late-Onset Cerebellar Ataxia"

_jcm, 2019, doi:10.3390/jcm8060789_

Round 1
Reviewer 1 Report
The manuscript is well written and complete. I only have minor concern regarding methodology:
Did the authors check genomic DNA variants ?
How did they exclude a genomic DNA variants ?
I also have minor concerning the impact of the message for the journal: do the authors think to suggest a muscle biopsy in any case of ataxia?
Author Response
Point 1
The manuscript is well written and complete. I only have minor concern regarding methodology:
Response
We are grateful for the reviewer 1's comments.
Points 2 and 3
Did the authors check genomic DNA variants ?
How did they exclude a genomic DNA variants ?
Response
Our patient was tested negative for spinocerebellar ataxia (SCA) 1, 2, 3, 6 and 7 as well as Friedreich's ataxia. We have included this information in the manuscript (Line 87-88, Page 2).
Point 4
I also have minor concerning the impact of the message for the journal: do the authors think to suggest a muscle biopsy in any case of ataxia?
Response
We thank Reviewer 1 for this helpful comment. We have revised the last paragraph as follows (Line 247-251, Page 7):
In summary, we recommend that muscle biopsy and full mitochondrial genome sequencing should be considered as part of the investigation for the adult-onset progressive cerebellar syndrome after excluding common acquired and genetic aetiologies, given de novo, rare pathogenic mtDNA pathogenic variants may present with atypical mitochondrial syndromes, as nicely illustrated by this case.
Reviewer 2 Report
The manuscript describes a novel pathogenic variant of of MT-CO2 identified from a patient with cerebellar ataxia and hearing loss. 13 other variants have been described in the literature. The manuscript is well, scientifically sound and also demonstrates that sequencing of mtDNA is an important tool to distinguish patients with autosomal recessive disorders from patients with a mtDNA disease.
Minor comments:
line 58: write "52 years".
Line 109: please provide sequence of amplification primers.
Figure 2: letters for panels are missing.
Line 163: write "patients".
Author Response
Point 1
The manuscript describes a novel pathogenic variant of of MT-CO2 identified from a patient with cerebellar ataxia and hearing loss. 13 other variants have been described in the literature. The manuscript is well, scientifically sound and also demonstrates that sequencing of mtDNA is an important tool to distinguish patients with autosomal recessive disorders from patients with a mtDNA disease.
Response
We are extremely grateful with the positive comments from Reviewer 2.
Minor comments:
Point 2
line 58: write "52 years".
Response:
Corrected.
Point 3
Line 109: please provide sequence of amplification primers.
Response:
We thank Reviewer 2 for this suggestion and have provided this information in the revised manuscript (Line 110 - 112, Page 3):
The entire mitochondrial genome was amplified using two overlapping long PCR amplicons, NC_012920.1:m.550-9839 (amplified using oligonucleotide primers m.550-569 and m.9839-9819) and NC_012920.1:m.9592-645 (amplified using oligonucleotide primers m. 9592-9611 and m.645-626)....
Point 4
Figure 2: letters for panels are missing.
Response
We apologise for the missing letters for panel A and B in Figure 2. This has been corrected.
Point 5
Line 163: write "patients".
Response
Corrected.